# Main Uncertainties in the RF Ultrasound Scanning Simulation of the Standard Ultrasound Phantoms

**DOI:** 10.3390/s21134420

**Published:** 2021-06-28

**Authors:** Monika Makūnaitė, Rytis Jurkonis, Arūnas Lukoševičius, Mindaugas Baranauskas

**Affiliations:** Biomedical Engineering Institute, Kaunas University of Technology, K. Baršausko Str. 59-455, LT-51423 Kaunas, Lithuania; rytis.jurkonis@ktu.lt (R.J.); arunas.lukosevicius@ktu.lt (A.L.); m.baranauskas@ktu.lt (M.B.)

**Keywords:** radio-frequency ultrasound, phased array, resolution phantom, echoscopy simulation, point spread function, image contrast

## Abstract

Ultrasound echoscopy technologies are continuously evolving towards new modalities including quantitative parameter imaging, elastography, 3D scanning, and others. The development and analysis of new methods and algorithms require an adequate digital simulation of radiofrequency (RF) signal transformations. The purpose of this paper is the quantitative evaluation of RF signal simulation uncertainties in resolution and contrast reproduction with the model of a phased array transducer. The method is based on three types of standard physical phantoms. Digital 3D models of those phantoms are composed of point scatterers representing the weak backscattering of the background material and stronger backscattering from inclusions. The simulation results of echoscopy with sector scanning transducer by Field II software are compared with the RF output of the Ultrasonix scanner after scanning standard phantoms with 2.5 MHz phased array. The quantitative comparison of axial, lateral, and elevation resolutions have shown uncertainties from 9 to 22% correspondingly. The echoscopy simulation with two densities of scatterers is compared with contrast phantom imaging on the backscattered RF signals and B-scan reconstructed image, showing that the main sources of uncertainties limiting the echoscopy RF signal simulation adequacy are an insufficient knowledge of the scanner and phantom’s parameters. The attempt made for the quantitative evaluation of simulation uncertainties shows both problems and the potential of echoscopy simulation in imaging technology developments. The analysis presented could be interesting for researchers developing quantitative ultrasound imaging and elastography technologies looking for simulated raw RF signals comparable to those obtained from real ultrasonic scanning.

## 1. Introduction

Ultrasound (US) imaging is the most widely used imaging modality in clinical practice due to its low cost, non-ionizing nature, noninvasiveness, and real-time imaging. US imaging technologies are continuously evolving. A lot of imaging advancements are related to the processing and parametrization of raw radio frequency (RF) echoscopy signals, opening new possibilities in tissue elastography, quantitative imaging, and the parametric mapping of quantitative tissue parameters valuable for differential diagnostics.

A powerful tool for further developments and modifications is the modeling and simulation of all echoscopy processes, especially in the RF domain. This in silico approach potentially allows simulating new concepts, methods, and implementations, which are unavailable in a clinical environment or even in laboratory experiments. However, its success decisively depends on the quantitatively evaluated adequacy of the simulation to the real echoscopy situation. Therefore, the simulation method and quantitative evaluation of involved simulation uncertainties are highly important. The general method of uncertainty quantification in simulation models proposed in [1] is based on consecutive forward uncertainty propagation (from the system under simulation towards the output of the system model) and the backward uncertainty quantification (consecutive evaluation uncertainty components from system model output to the real system). In the case of US echoscopy, the simulation uncertainty components are related to models of multi-element transducer and beamforming, US wave propagation and 3D field, backscattering, and speckle formation in nonhomogeneous tissue, transmit-receive mode, RF data acquisition, and preprocessing. Therefore, echoscopy deals with complex interlinked modeling and simulation tasks and requires reliable simulation methods and software tools.

An adequate modeling of US wave propagation in the biological tissue involves solving the nonlinear Kuznetsov–Zabolotskaya–Khokhlov (KZK) equation which can also be derived as the parabolic approximation to the Westervelt equation [2]. The direct solution of the equations is quite complicated and limited to directional wave propagation. From the KZK equation, computational resource-saving K-wave model and angular spectrum approach (ASA) were derived, which are based on some simplified assumptions of linearity and space invariance of the point spread function (PSF) [3,4,5]. These methods balancing between adequacy and needed computational resources are used for US field modeling and simulation of the wave propagation in tissue [6], but not directly applicable for the transmit-receive scanning of scattering media by multielement transducer arrays.

Popular simulation packages include CREANUIS [7] (based on KZK equation) for nonlinear propagation and simulation of harmonic imaging and Field II [8] (based on Huygens principle and hybrid digital-analytic method) which gained a wide application since the publication of the model in 1991 [9] and the software package in 1996 [10]. The popularity of Field II lies in its considerable flexibility to accommodate a wide range of transducers, beamforming options, and imaging possibilities. It is still a powerful and evolving tool for the simulation of various scanning and focusing modalities as well as whole echoscopy mechanism including scattering in the tissues and organs [11]. Scanning by a linear array transducer simulated by both Field II (as a reference) and ASA shows that Field II is effective in the simulation of the US field generated by the transducer, while ASA is effective in nonlinear propagation [12]. A comparison of several new resources-saving simulation tools for fast simulation of US data has shown [13] that methods based on the principle of convolving a set of point scatterers with a PSF do not simulate the process of US field and beam formation and interaction in echo mode in contrast to Field II. The simulations of plane wave images made with Field II using wire phantom and a tissue-mimicking phantom having anechoic cysts [14] have shown a high value of simulation which could be applied for the optimization of imaging.

Digital models of tissues and organs are also a component of in silico echoscopy and a source of simulation uncertainty. Nonhomogeneous tissue is usually simulated by discrete digital phantoms based on the small scatterers distributed in space. The proper setting of scatterer maps is essential for the realistic resulting image of ultrasonic wave interference (speckle) pattern [15]. The mean scatterer spacing (MCS) for different tissues was reviewed in [16] showing the potential to simulate tissue microstructure by fitting the MCS parameter. Scatterer maps also could be generated from real images by solving the inverse-problem of US speckle formation [17].

Although there are a lot of Field II applications described in many articles, only a few publications deal with the quantitative comparison of simulation and experiment evaluating uncertainties. The adjustment of Field II results was attempted in [18] by including a model of one-dimensional transducer impulse response and hydrophone experiment provided to measure the field pressure for the comparison with Field II simulation. Point reflectors and RF signals simulated using Field II have shown reflector contrast and signal to noise ratio estimates [19]. There, PSF simulation is used for reducing the amount of RF data samples transferred for visualization, without a quantitative assessment of PSF simulation adequacy. The brief comparison of Field II simulation and experimental results provided on wire images was presented in [20,21] and [22] without a quantitative assessment of adequacy.

Most simulation results are for the simpler case of a linear scanning transducer arrays. Articles mainly use Field II simulation results as a reference for the analysis of some US imaging development without a prior assessment of simulation adequacy.

The methodology for simulation of 3D echocardiographic sequences presented in [23] is based on a convolution of phased array PSF with point scatterers. This approach makes calculations much faster than the more accurate simulation by Field II [24,25], but space invariant PSF is taken from theoretical assumptions and PSF adequacy and all the simulations are not verified experimentally. A simulation of space variant phased array PSF with Field II is provided [26], but no experimental verification of PSF parameters on phantom, and no data about resolution in elevation direction are given.

By the Field II simulation of phased 1.75D array, it was shown that beam width in elevation direction (or scanning slice thickness) has a significant influence in B-image contrast reproduction [27]. This was shown by experimental contrast verification with a phantom of spherical inclusions, but no verification of quantitative beamwidth estimates in elevation direction was provided. The experimental analysis of elevation beamwidth profiles made for the freehand ultrasound calibration purposes showed the importance of scanning slice thickness for linear-array transducer calibration accuracy [28], but no results for phased array were provided and no digital simulations. The improvement of the ultrasound transducer radiation beam profile was verified by hydrophone scanning of the pressure field [29], but not simulated. Thus, the quantitative comparison of phase array simulation with an experiment including an analysis of resolution in the elevation direction remains unsolved.

The problem of simulation adequacy is especially crucial for US elastography (USE), where tissue displacements could be extremely small and resolution in strain estimation is important [30]. For example, Field II could be used not only for static US images but also for pulsating carotid artery simulation [31]. It is also effective in combination with a convolution-based method [32] which shows the potential of 3D moving scatterers and tissue deformation modeling using the finite element method (FEM) and the corresponding pre-/post-deformation RF signals comparison using Field II [33,34]. Despite the very wide use of Field II simulation software, the remaining problem for the development of new imaging modalities including USE is a quantification of simulation uncertainty, especially for sector scanning. An adequate simulation of tissue deformations would open a way for the more rapid development of USE, which enables differential tissue diagnostic [35,36] including the use of the endogenous motion of tissue induced by cardiovascular activity, where experiments are limited because of technical and principal constraints [37].

Since digital simulation is increasingly important for US imaging and elastography developments, a quantitative assessment of uncertainties involved by all interrelated components of the simulation process as well as revealing of the possible factors causing those uncertainties remains a problem.

The aim of this paper is a quantitative evaluation of the main resolution and contrast related uncertainties arising in digital US sector scanning simulation by Field II and comparing them with physical phantom’s scanning results leading to the conclusions on quantitative uncertainties of PSF, resolution in the elevation plan, and image contrast reconstruction using standard phantoms. The contribution of our study is in the analysis of underestimated sector scanning simulation uncertainties.

## 2. Materials and Methods

### 2.1. Simulation Uncertainty Analysis Concept

The principle of simulation uncertainty analysis is depicted in Figure 1. The resolution in three (axial, lateral, and elevation) directions and image contrast reconstruction parameters were estimated using standard physical phantoms and corresponding digital models. A real scanner with an RF signal registration facility was used for the scanning of dedicated phantoms and the results were quantitatively compared with the Field II RF signal simulation results and uncertainty estimates provided. For correct comparison of results, both sets of RF signals (experimental and simulated) were used to generate B-scan images by common algorithm, thus avoiding the influence of signal-to-image scanner functionalities on the results. We used 3D digital phantoms for the simulation of standard physical phantoms and analysis of arising simulation uncertainties. RF signals were simulated in sector scanning mode. The resolution in scanning plane by quantitative PSF analysis was supplemented by the resolution estimate in the elevation plane perpendicular to the scanning plane. Backscattered RF signals and contrast reproduction in the simulation were shown. Following the aim to disclose an uncertainty of real echoscopy simulation, attention was mostly paid to fitting the model parameters to the experiment, not vice versa. We exclude fitting experiment parameters to the modeling possibilities. In addition, aiming to eliminate scanner imaging functionality from the comparison of simulation and experiment results, B-scan images were calculated from raw RF signals both in the simulation and experiment.

The results are provided in illustrations and tables of quantitative estimates leading to the analysis of complex sources of simulation uncertainties.

### 2.2. Physical Resolution, Contrast and Slice Thickness Phantoms

For the echoscopy experiments, three physical phantoms were used: ATS Laboratories Inc. (Brawley, CA, USA), i.e., Model 549 (general and small parts phantom) [38], Model 532A (contrast resolution phantom) [39], and Model 538NH (slice thickness phantom) [40]. Part of the Model 549 phantom was used in a kind of wire phantom for axial and lateral resolution evaluation (see Figure 2a marked part), part of the Model 532A phantom was used for contrast resolution detection (see Figure 2b marked part), while Model 538H was used for resolution estimate in the elevation plane of transducer (see Figure 2c marked part).

According to technical passports of the phantoms, point targets are the physical wires—monofilament Nylon wires with a diameter of 0.05 mm while the physical position tolerance declared is ±0.1 mm [38]. The depth of the first wire in the preselected scanning area (see Figure 2a marked part) was measured with a physical scanner, using the speed of sound set to 1540 m/s. It was found that the depth of the first wire is 44 mm.

In the physical contrast phantom, the contrast of surface nearest inclusion was +12 dB regarding background. The manufacturer’s declared standard deviation of contrast values is ±5%. According to serial numbers of phantoms, the declared individual speed of sound—1453 m/s, 1452 m/s 1451 m/s in the urethane rubber-based tissue of physical wire, contrast phantoms, and slice thickness phantom mimicked the materials accordingly, however, these phantoms were constructed to be used with the speed of sound with 1540 m/s. Attenuation—0.501 dB/cm/MHz@3.5 MHz, 0.514 dB/cm/MHz@3.5 MHz, and 0.497 dB/cm/MHz@3.5 MHz in the physical wire, contrast phantoms, and slice thickness phantom, accordingly. For all these urethane rubber-based phantoms the acoustic properties were measured at +23 °C by ATS Laboratories. The inevitable deviations of the passport parameters of phantoms influence the simulation uncertainty experimental validation.

### 2.3. Digital Phantoms for Resolution, Contrast and Slice Thickness Simulations

For the comparison of scanning results obtained with physical phantoms, appropriate digital phantoms were designed using a spatial distribution of acoustic scatterers.

The number of scatterers in the volume of digital phantoms was set proportionally to a measured physical phased array transducer resolution cell and the volume of the phantom model. Scatterers were randomly distributed in the model volume with amplitudes distributed by Gaussian function with mean 0 and standard deviation 1. To achieve a fully developed speckle pattern in US image calculated by Field II software chosen as a simulation tool, we used 10 scatterers per resolution cell (SPRC) [21,41], which is calculated by [42]:(1)$$resolution\;cell=\lambda \cdot N\cdot FWHM_{lat}\cdot FWHM_{azi},$$
where *λ* is the wavelength, *N* is the number of cycles in the pulse, *FWHM_lat_* and *FWHM_azi_* are full width at half maximum (FWHM) laterally and axially, respective of the PSF at the focus point. In our case, the 2.52 MHz multielement transducer resolution cell was:(2)$$resolution\;cell=\frac{1540\;m/s}{2.52\cdot 10^{6}\;\mathrm{Hz}}\cdot 2\cdot 0.59\cdot 10^{-3}\;m\cdot 2.66\cdot 10^{-3}\;m\;\approx \;1.92\cdot 10^{-9}{\;m}^{3}=1.92{\;\mathrm{mm}}^{3}.$$

The total number of scatterers in the digital model’s tissue-mimicking material (TMM), which was used in the simulation, was calculated by:(3)$$number\;of\;scatterers=10\cdot \frac{volume\;of\;model}{resolution\;cell}.$$

To have the adequate amplitudes of reflected simulated signal, amplitudes of the scatterers which are modeling phantom inclusions (point targets, wire targets, contrast cylinders, and inclined plane), were scaled relative to the scattering amplitudes of scatterers modeling TMM:(4)$$PSA=max\left(BSA\right)\cdot 10^{\left(\frac{dB_{PS}}{20}\right)},$$
where *PSA*—amplitude of the scatterers modeling inclusions, *BSA*—background scatterers amplitude, and *dB_PS_*—scattering contrast in decibels according to the B-scan image of the scanned objects or specification data of physical phantom.

**Wire phantom.** Since phantom used for resolution and PSF estimation contains monofilament wires with a 0.05 mm diameter [38], we designed an appropriate digital wire phantom with wires of the same diameter at particular depths (see Figure 3a). However, by correct definition, PSF is a result of US wave reflection from the point scatterer, not from a wire. Therefore, for comparison of results, the digital phantom containing point scatterers located at the same positions was designed as well (see Figure 3b). The positions of seven scatterers in both versions of phantoms were set from 44 mm every 10 mm to 104 mm away from the transducer in the axial direction.

**Contrast phantom.** The digital contrast phantom (see Figure 3c) was designed to simulate physical phantom Model 532A (see Figure 2b). Phantom inclusion was modeled as 4- and 8-mm diameter cylinders with +12 dB contrast regarding to background calculated by (4) formula and density distributions as described above. Some experiments were provided to evaluate the influence of the increased number of scatterers (from 10 to 15) per resolution cell on the results of the simulation.

**Inclined plane phantom.** B-scan slice thickness or beamwidth in the elevation plane were evaluated according to the method pioneered by Goldstein and Madrazo [43] and updated later [44]. It could be relatively concerned as a third geometrical dimension of PSF. An inclined plane phantom with a 45° inclined scattering plane [40] was used for the experimental estimate of evaluation resolution while digital slice thickness phantom—in simulation. Digital inclined plane phantom (see Figure 3d) for the simulation of slice thickness in the elevation plane (transverse to the plane of scanning) was designed in accordance with the physical phantom Model 538NH (see Figure 2c). The model has a 45-degree inclined scattering plane simulated as a layer of randomly distributed scatterers. Because there was no information about the plane characteristics, i.e., thickness, of the physical inclined plane phantom, we decided that the thickness of the layer would be the same as monofilament wires diameter in physical wire phantom i.e., 0.05 mm.

The configuration of all digital phantoms together with the position of the US scanning transducer marked is presented in Figure 3 (in the case that the wire phantom transducer scanning plane is perpendiclular to the wires).

Since TMM in phantoms has frequency-dependent attenuation like real tissues, this feature was also considered in the simulation. Ultrasound attenuation parameters were taken from the phantom’s documentation.

### 2.4. Physical Scanner

The experimental scanning using all phantoms was provided for comparison with a digital simulation. It was done by a research-dedicated US scanner Ultrasonix Sonix Touch (Analogic Ultrasound, Richmond, BC, Canada), equipped with an SA4-2/24 planar phased array transducer. The scanner allows the acquisition of digitally beamformed US RF signals. The main parameters of US RF signal digitization and acoustic scanning were as follows: 16 bits analog-to-digital converter, 40 MHz sampling frequency, 2.52 MHz frequency of US waves, the number of transducer elements was 64, the number of post-beamformed scanning lines was set to 131, and the echoscopy depth and transmit focal depth were set to 11 cm and 7 cm accordingly. The phased array transducer sector composed a 63-degree angle, which ensures a higher frame rate to capture pulsing tissues dynamics in vivo in future studies. The speed of sound in the scanner and for the processing of the US RF signals was set to 1540 m/s.

The comparison of the experimental scanning results obtained by the physical scanner with results of digital simulation is complicated because some internal scanner parameters are unknown while some features of the scanner cannot be adequately simulated by Field II software. To partially overcome this problem, the advanced features of scanning (such as compensation of RF signal frequency downshift caused by attenuation, also frequency sliding, used in signal demodulation, harmonic imaging mode were switched off. The known parameters under control were used. Uncertainties in comparison could also be introduced by the imaging system of the scanner in the path of transformation of RF signals into the B-image on the monitor screen. Therefore, we used the raw RF signals of scanning lines registered from the scanner for creating a B-image by the external computer.

### 2.5. Digital Simulation of the Scanning (Virtual Scanning)

A digital scanner based on a phased array transducer was simulated using the Field II package for Matlab [45]. To reach adequacy between experiments and simulation results, we identified available parameters of physical scanner and ultrasound transducer and used them in the Field II program. The height and width of elements, kerf, pitch, number of elements, and elevation lens focus were found in the Transducer Specification Sheet [46]. Speed of sound, transmit focal depth, and element excitation was identified from the scanner settings. Assumed frequency-dependent attenuation data was taken from the phantoms data sheet [38,39,40]. The initial values of Field II parameters together with transducer and scanner values used to simulate scanning with phased array transducer, are listed in Table 1.

In parallel with RF data, the Ultrasonix Sonix Touch scanner stores the default header where the RF signals sampling frequency is given. The Ultrasonix Sonix Touch scanner is a research dedicated system, therefore, more data is made available from the clinical scanners. XML files are saved by the scanner automatically with each B-scan image and other types of sonography data. In the XML file, extra technical parameters of sonography are made available to researchers. The array transducer center frequency was found in the XML file, which was saved together with preselected types of saved data that are desired to be stored in the scanner disk.

To increase the accuracy of the simulation, e.g., elevation lens focus, apodization in the YZ plane, the physical elements must be divided into the *sub_X_* by *sub_Y_* mathematical elements. This could be done by changing the number of physical element sub-division in x and y directions (XZ plane and YZ plane correspondingly) and investigating the influence on the pressure field of the simulated transducer. Subdivision in X axis direction was *sub_X_* = 1, because physical elements are of small (0.24 mm) width. The X axis describes the longer dimension of the phased array (see Figure 3).

Field II enables modeling of the non-uniform distribution of generated US signal over the transducer element surface. Therefore, a choice of the number of element sub-divisions in x and y directions was made regarding side lobe level and computational resources needed. The calculations of pressure field were provided when subdivision numbers *sub_X_* and *sub_Y_* were from 1 to 5 and from 1 to 20 accordingly. Element sub-division in x and y directions was considered sufficient when side lobes were not observed in the pressure field in XZ and YZ planes correspondingly. After qualitative visual inspection of the simulated pressure fields in XZ and YZ planes under different combinations of element sub-division, *sub_X_* = 1 and *sub_Y_* = 13 were selected as sub-optimal values for virtual scanning with Field II.

To have the same formation of the B-scan image in both physical and digital scanning, we chose to load registered (i.e., physical scanning) and modeled (i.e., digital scanning) beamformed US RF signals into MATLAB. Then B-scan formation consists of the following steps: 1. RF signals envelope detection; 2. log-compression from 0 to −73 dB.

In a digital simulation of scanning simultaneously are concerned models of the scanner, transducer, and phantom. Modeling inadequacies of those models could potentially be sources of uncertainty. Numerous parameters of the physical scanner/transducer shown in Table 1 have inevitable uncertainties which could be unknown fully and incompatible with the Field II parameters; as well as the uncertainty of reference data declared in the phantom documentation. One of the unknown parameters is an apodization function of the transducer aperture in the plane perpendicular to the scanning plane. It is dependent on the technology of the piezoelement, matching layers, and lens and is hardly measurable and specified, although it has an influence on the resolution. Therefore, the approximation of this apodization by division in certain parts is not exact due to the lack of an explicit description. Element sub-division in y-direction is therefore indefinite. The other example of unknown parameters is the parameters of US pulse emitted by a single element of phased array transducer. Other unknown parameters of cross-talking between transducer elements, dispersion of amplitudes, beamforming errors, and others also influence the final simulation uncertainties Therefore, the full adequacy of simulation evidently is not achievable; rather, determining the level and possible causes of uncertainties are the goals of the analysis. We keep the concept of digital modeling and scanning simulation by considering only available reference data from physical phantoms and scanner/transducer. First, we set all available transducer, scanner, and phantoms parameters in the Field II model. More comprehensive simulations were repeatedly provided for the wire phantom, inclined plane phantom, and contrast phantom. After simulation with the initial parameters, we compared the experimental and digital simulation results and evaluated compliance. The analysis of compliance showed uncertainties of simulation which could be minimized by the additional fitting of digital model and simulation parameters. It should be noted that in the case of additional fitting, the simulation parameters giving the best experiment-simulation matching could differ from formally available physical scanner parameters. However, this is out of the scope of the present paper.

## 3. Results

### 3.1. PSF from Experimental Data and Simulation

FWHM was used as a resolution parameter that determines quantitatively the axial, lateral, and elevation width of PSF of both measured physical and simulated digital phantoms. The simulation of wire phantom using the scanning parameters, shown in Table 1, was compared with experimental scanning results and presented in Figure 4. PSF axial positions are compared using a common axis of depth (see Figure 4a). The axial and lateral profiles of experimental and simulated PSFs in the case of wire phantoms are compared using common normalized axes. The positions of simulated peaks of PSF profiles are all observed centered at zero of the axes named as PSF width, while experimental PSF profiles are slightly deviating from this position. A summary of quantitative FWHM parameter comparison between measurement and simulation in case of wire phantom (see Figure 2a) is presented in Table 2.

FWHM values of PSF for seven equidistant phantom wires presented a comparison between experimental echoscopy and digital simulation (see Table 2) shows that the average error of axial and lateral resolution is 8.8 and 8.3%, accordingly.

To evaluate the suitability of the wire phantom for the estimation of echoscopy resolution using PSF (which by definition should be measured using not a wire but a point scatterer), we simulated a scanning of the digital phantom containing point scatterers (see Figure 3b) and compared results with real echoscopy of wire phantom (see Figure 2a). The results presented in Figure 5 and Table 3 show comparable results as with wire phantoms. The average errors of axial and lateral resolution are 4.6 and 5.2%, accordingly (see Table 3).

The results presented in Table 2 and Table 3 shows that axial dimensions of simulated PSF using point scatterers are at a smaller average discrepancy from the case of wire scatterers (4.6% vs. 8.8% accordingly). In addition, at all distances, the axial dimensions of the simulated PSF of both scatterer types are bigger than the practical estimates.

The lateral dimension of PSF was found to be less influenced by the type of scatterers than the axial dimension, showing an average discrepancy of 8.3 and 5.2%, accordingly. At the focal region (from 54 to 85 mm) both types of scatterers show bigger simulated lateral PSF dimension than a practical estimate. While outside of the focal region the field models of both types outperform the practical estimates of PSF lateral width (see negative errors in Table 2 and Table 3), the PSF lateral and axial dimensions of the point target model are in better agreement with the practical estimates as the average discrepancies of only 4.6% and 5.2% were obtained (see Table 1). As expected, both axial and lateral dimensions of PSF calculated from a single point are narrower than simulated from wire scatterers. Some differences caused by scattering from point and wire scattering could also be seen in the PSF shape outside of the main peak. A two-dimensional comparison of practical and simulated PSF is provided in the Section 3.3.

### 3.2. Experimental and Simulated Pulse Waveform and Frequency Spectra

The adequacy of the simulation depends on the conformance of the simulated ultrasound pulse shape to the experimentally measured one. The emitted ultrasound pulse model is 2 periods of 2.52 MHz oscillation modulated by the Hanning function. Figure 6 presents the pulse shape and frequency spectra of reflection from wire target located 74 mm from the transducer simulated by Field II and experimentally measured in the same position with the phantom. The comparison shows a good agreement since the results are influenced by ultrasound diffraction and other propagation effects.

### 3.3. PSF Two-Dimensional Comparison for Wire and Point Targets

Since PSF is the most important measure of resolution, a more detailed analysis in two dimensions was provided (see Figure 7). The axial dimension of PSF is influenced by the pulse duration and spectrum which is quite similar both in simulation and experiment (see Figure 6) and is much smaller than the lateral one. At a focal distance *z* = 74 mm, the lateral dimension is smaller than at a far-field *z* = 104 mm, but simulation and experimental results presented in Figure 7 have a good agreement. From the two-dimensional representation of PSF, we see some peculiarities of simulation results: an asymmetrical profile in axial direction—in the axial direction the PSF has a steeper slope from the transducer side if compared to the far-field side; also, the fact that the peaking coordinate of PSF is closer to transducer than the midpoint of the contour of 0.5 level of PSF.

Some observed differences can be caused by the directivity of simulated and practical transducer beam in the elevation direction. The finite beamwidth in elevation direction supposedly increases the backscattering from the finite length of the wire target so it could result in a wider axial dimension of PSF. Directivity in elevation direction was evaluated additionally using scattering plane oriented at 45° angle to the scan plane, as described in Section 3.4.

### 3.4. Experimental and Simulated Beamwidth in the Elevation Plane

The resolution of US imaging is defined not only by PSF having axial and lateral dimensions measured in the scanning plane but also by the elevation resolution, to be measured in the plane perpendicular to the scanning one. Inclined plane (slice thickness) phantom with a 45° inclined scattering plane [40] (see Figure 2c) was used for the experimental estimate of evaluation resolution while digital slice thickness phantom (see Figure 3d) was used in the simulation. The distance of the B-scan central line intersection with the scattering plane was 70 mm as observed in the scanner monitor. Using Field II software and the 45° inclined scattering plane digital phantom (see Figure 3d) was simulated RF signals at the scanner output and corresponding B-scan images composed. The experimental and simulated B-scan images of this case are presented in Figure 8, while the envelopes of the central scan lines obtained from the inclined plane of physical and digital slice phantoms—at Figure 9a,b accordingly.

The elevation resolution of physical echoscopy and simulation (see Figure 8 and Figure 9) is in qualitatively good agreement. In simulated and practical B-images (see Figure 8) and in envelopes of central scanning line along axial scanning direction (see Figure 9) specular pattern is seen. In sonography, the PSF width in the elevation plane usually is represented on the axis of axial distance [43,44]. According to estimation methods, the widths of PSF in axial and elevation directions are inter-related. In our case (Figure 9) the FWHM of the beam in the elevation plane is 2.5 mm in physical and 3.2 mm in simulated beams, i.e., the simulated beamwidth appears wider than the experimental one. Supposedly, this is due to the insufficient simulation accuracy of transducer aperture apodization in the elevation direction.

### 3.5. Experimental and Simulated B-Scan Images

The next stage of echoscopy simulations is based on wave scattering from randomly distributed scattering centers representing TMM. Experimental and digital wire phantoms were used. B-scan images and central scanning line envelopes are compared in Figure 10. It is important to note that the algorithm converting US RF signals to the B-scan image was common in both cases—of experimental and simulation RF signals. So, the proprietary image transformations executed by the scanner are excluded from the comparisons. The main difference is seen in the scattering from wire-targets and background TMM since the energy of ultrasound waves reflected from physical phantom wires is unknown and difficult to simulate adequately. Therefore, the ratio between signals from wires and background is difficult to reproduce accurately. The difficulty lies also in the fact that scattering parameters in standard TMM phantom are not identified in the available documentation, but the additional fitting of those parameters was out of the scope of the present paper.

### 3.6. Image Contrast and US RF Signals

A correct reconstruction of scattering contrast in scanning image and in US RF signals together with the spatial resolution is an important scanning quality parameter. The scattering of US waves is modeled with specific distribution of specific scatters in background material. The commercial contrast phantom Model 532A has cylindrical inclusions with higher scattering (contrast +12 dB) in relation to scattering from background TMM. Physical echoscopy of contrast phantom Model 532A (see Figure 2b) was simulated by a digital contrast phantom (see Figure 3c). The scanning transducer was placed nearest to the inclusions of Model 532A phantom (as in Figure 3c). Two digital phantom versions of physical contrast inclusions were calculated using two different scatterer densities: 10 SPRC as recommended by modeling literature [21] and 15 SPRC—for comparison. Simulated B-scan images are presented in Figure 11b,c accordingly.

Figure 11 together with B-scan sub-images shows RF signals from physical and digital contrast phantoms. RF signals (see Figure 11d–f) represent the scattering pattern in background material and contrast inclusion along one scanning line at (−1) mm lateral position. For a quantitative evaluation of contrast, image regions were outlined (see Figure 11a–c) to calculate mean B-scan intensities (mean of B-scan image pixels’ intensities in 7 × 7 mm regions): in background IB and inclusion II. The contrast was calculated as the difference of mean intensities obtained in regions of background and inclusion. From outlined regions of experimental B-scan image in Figure 11a calculated intensity values of IB = −19.2 ± 0.7 dB and II = −10.2 ± 0.5 dB results into contrast of +9 dB. The contrast of +9 dB derived from the experiment shows an underestimation of the reference value (+12 dB) found in the Model 532A description.

The same estimation of contrast was made for images simulated with digital contrast phantoms (see Figure 11b,c). Intensities in the regions of simulated image with 10 SPRC were IB = −25.2 ± 0.5 dB and II = −8.2 ± 0.5 dB, resulting in the contrast of +17 dB. In case of model with 15 SPRC the corresponding estimates were IB = −24.5 ± 0.5, II = −8.1 ± 0.6 dB and resulting contrast +16 dB. The approach of scattering contrast modeling by fitting only scatterers density by formula 4 is resulting in overestimation and appears not adequate. Unfortunately, the documentation of Model 532A phantom does not contain adequate data about scatterers in TMM as well as no information about frequency dependence of scattering.

RF signal morphology examples are presented in Figure 11d–f. It is important that the simulated RF signal is an integral result of all echoscopy components: multi-element array transducer, beamforming, wave diffraction, focusing, aperture apodization, and backscattering in TMM. The latter phenomena, backscattering, is highly dependent on scatterers’ distribution in phantoms.

## 4. Discussion

The echoscopy simulation by Field II was chosen because it is a software that models multielement array transducer in transmit-receive mode and takes into account focusing, apodization, and wave attenuation mechanisms simultaneously and generates US RF signals received from arbitrary 3D distributions of point scatterers. Despite the wide use of Field II in echoscopy simulations a quantitative experimental estimation of simulation results and their uncertainties still is an undermined problem and hence important for those using simulated data for the development of new US imaging and elastography technologies.

The use of standard physical phantoms dedicated for testing medical diagnostic scanners has an advantage because standard phantoms have individual passports with main parameters defined in the documentation. However, as the presented results illustrate, declared parameters have both declared tolerance and undefined discrepancies, which reflect simulation uncertainties. A possibility to accumulate experimentally the raw RF signals from each scanning line allowed transferring raw scanning data to the computer where reconstructions of B-scans were made for comparison with simulations thus overcoming quite sophisticated and not completely known image formation procedures in the scanner itself. Quantitative FWHM measures of PSF were presented in axial, lateral, and elevation directions. FWHM estimates compared in Table 2 shows that in all depth positions FWHM estimates in the axial direction are according to practical estimates with a discrepancy of less than 10%. While the discrepancy in lateral direction deviates with depth and exceeds 30% in one depth before focusing region. PSF views in Figure 4 show that the simulation results of an axial view of normalized PSF coincide better with the experiment than lateral one, which is supposedly influenced by difficulties to account in simulation parameters of apodization, focusing, and other ultrasound beamforming measures taken by the scanner to equalize the lateral resolution over the depth. (Adjustment of Field II simulation with the scanner is limited to the available set of parameters, see Table 1). The mismatch of lateral positions of simulated and experimental PSF is up to 1 mm. Discrepancies could be accounted to the internal deformations of urethane rubber with embedded scattering wires as the phantom is already 10 years past manufacture. The results also show some small discrepancy in the axial position of reflections from equidistant wires—both experimental and simulated results did not meet exactly distances presented in the phantom passport with tolerance ±0.1 mm as well as ultrasound velocity. The position error not exceeding 0.3 mm shows a rather good agreement of simulated data to the experiment with Model 549 phantom.

The results of the simulation of wire target phantom by both wire and point scatterer target digital phantoms answer the question of whether the point targets are suitable for the simulation of standard wire target phantoms. The data presented in Table 2 and Table 3 show that simulating of wire target by the point target model gives a 4.6% average axial resolution error to compared with the experiment and according to the error for simulation by point target gives 8.8%. Comparing scattering aside of the main-lobe in a point model versus wire model, the point model exhibits relatively stronger side-scattering of the farthest objects (see Figure 5). Evidently, the wire is insonated by an ultrasonic beam of finite width in elevation direction and the additional backscattered echo from wire target is accumulated on transducer aperture, forming the average or smoothed result of the speckle. In the case of echo from a single point target, the beam of finite width in elevation direction makes no accumulation, resulting in no smoothing on transducer aperture. Therefore, a weaker speckle pattern is present on both sides of the axial main lobe calculated from the wire target (see in Figure 4), and the stronger is in the case of the point target (see Figure 5). This may be because PSF calculated from point and wire models are separately normalized, possibly exhibiting more noise in case weaker energy is scattered from the point target. It is interesting to mention that the simulation comparison of digital wire phantom echoscopy and digital point scatterers phantom (see Table 3) does not show big differences, especially in lateral resolution. One can conclude that a wire phantom is practically suitable for PSF estimations when practically it is difficult to produce a single-point scatterer phantom. The point model of PSF simulation is also favorable for quicker computing.

The experimental and simulated B-scan images of wire phantom show rather good agreement, the only level of the speckle noise level in the background differs—the simulation shows it higher (see Figure 10). Supposedly, this is related to the adequacy of speckle reproduction by the distribution of point scatterers in the volume of the digital phantom. We used scattering parameters of simulated scatterers without particular fitting to those of the physical phantom as the parameters of scatterers inside of standard ATS phantoms were unavailable in the specification.

Since US echoscopy is a process in space and time, resolution in space, not only in the scanning plane matters. The results of resolution simulation in elevation plane (perpendicular to the scanning plane) in Figure 8 and Figure 9 show a rather good agreement with real echoscopy of specialized Model 538NH phantom. The elevation resolution is related to axial resolution since it is another projection of the 3D space region disturbed by the US pulse. A scattering pattern in the B-scan image is observed at the same axial distance independently of scatterer location in respect of beam width in elevation direction [47]. The observed peculiarities of PSF models with wire and point targets we relate to the different contributions of scatters out from the scanning plane.

The results of contrast phantom simulation are related to the reconstruction of established speckle pattern—the result of US wave’s interference from regions with different echogenicity. The exact morphology reproduction in simulated backscattered RF signals is difficult because of the lack of information about scatterers in physical phantoms. Usually, some scattering powder is used in manufacturing standard phantoms to cause backscattering of different intensities, but the acoustical parameters of scattering useful for simulation remain unknown in commercial phantoms. Backscattering and attenuation of US strongly and non-linearly depend on frequency. However, this information is given only by the approximate linear dependence at the particular frequency for ATS phantoms. The lack of comprehensive data about physical phantoms currently limits a more accurate simulation but the results of simulated RF signals and contrast reconstruction are quite promising. A comparison with results in [21] shows that the presented simulation could be useful for the development of quantitative imaging algorithms for echoscopy.

A comparison of the quantitative results with other researchers in Field II simulation of phased array scanning is quite difficult because of a limited number of articles dealing with experimental verification of such simulations. The phased array lateral PSF width at 60 mm depth was simulated with Field II for 5 MHz frequency by [48] and FWHM was obtained 1.4 mm without an experimental verification of this result, since experiments were provided with a curvilinear array transducer. Our estimates of lateral PSF width at a focal distance of 65 mm for 2.52 MHz frequency (which is twice lower) are in the 2.4–2.9 mm range both for simulation and for the experiment. Field II simulations performed for a 3.5 MHz phased array with focal depth at 30 mm and spatial resolution was quantified by measuring lateral FWHM width in resolution target phantom [49]. Simulated resolution (FWHM) with the phased array at focal depth was 1.01 mm lateral and 0.6 mm axial. However, again, the simulations were not verified since compared with experimental results of curvilinear array transducer of 3.1 MHz. Simulated lateral FWHM which is found minimal at a focal distance is almost double wider in the near field (1.5 mm at 1cm) or far field (2.41 mm at 5 cm). Field II calculation results were obtained for point reflector at a distance of 2d, where d is the length of phased array with 64 elements [50]. At a frequency 3.5 MHz and distance of 2d the FWHM of the lateral main lobe was obtained about 2 degrees of angle; without experimental verification of this result. Although a direct comparison of our results is difficult, they are comparable and contribute to the experimental verification of the simulations.

An experimental investigation of scanner’s resolution in three space directions for 3 MHz phased array transducers was presented in Long et al. [51]. The individual measurements as a function of depth agree with expectation: elevational resolution is worst (minimum 2 mm) and varies the most versus depth, with a distinct hourglass profile; the axial resolution is best and remains roughly constant (around 1 mm) versus depth; the lateral resolution is between the other two and demonstrates modest changes versus depth. Although no simulation is provided in this article, our experimental results are fully compatible.

The limitations of the present research are related to the possibilities to know all parameters of the ultrasound transducer, beamforming, and standard phantoms as well as with the ability to control needed parameters of Field II software. An example of an unknown parameter is the emitted pulse from the single element of phased array transducer. The only theoretical description of pulse that is used for excitation of elements we describe in Table 1: the Hanning-modulated sinusoid of two-cycle and transducer center frequency 2.52 MHz. However, such characterization does not ensure precise knowledge of the spectra of emitted ultrasound pulses. The submodel of the emitted pulse could change the current implemented theoretical excitation considering the electromechanical response of the piezoelement including impedance matching layers. The rather limited simulation accuracy of the lateral resolution is due to the lack of available information about the peculiarities of focusing and apodization both in transmit and receive. Apodization, especially in elevation direction for 1D apertures, is practically defined by transducer technology and is difficult to specify. The simulation and experimental resolution results should be better if the simultaneous fitting of parameters is provided. But we did not provide “backward fitting” of scanner parameters to the parameters of Field II, which would give better results. The limitation of the study lies in the use of a quite standard distribution and density (10 and 15 scatterers per resolution cell) of scatterers mimicking the scattering in physical phantoms. Commercial ATS phantoms are insufficiently specified concerning complicated and interrelated properties of backscattering required in quantitative ultrasound imaging research. Real tissue—US interaction mechanisms based on interference of backscattered waves in different tissues are more complicated and need further development. Some scattering discrepancies between experiment and simulation observed in Figure 10a,b are due to rather simple reasons—Gaussian distribution of scatterers amplitudes which is approximate modeling of complex non-homogeneous media with backscattering. Other distributions could give more adequate simulation results, especially regarding contrast and RF signal morphology. Reference phantoms for quantitative ultrasound imaging are not yet available commercially. They are still in the research phase: for example, phantoms for ultrasound waves backscattering [52,53] for waves backscattering and attenuation [54].

The simulation of echoscopy is a multitasking issue because the outcome of real scanner—RF signals and quality of the B-scan image is a result of many transformations including those induced by the transducer, propagation in non-homogeneous media (tissue), focusing, apodization, filtering, and image presentation. In each stage of those transformations, we face simulation uncertainties related to insufficient knowledge of parameters to be used in the simulation model. The present article is an attempt to evaluate and discuss the main simulation uncertainties which are important for the use of simulated data for the development of imaging technologies.

## 5. Conclusions

Simulated signals in contrast to clinical ones can quantitatively imitate a wide range of ultrasound interactions, which in clinical echoscopy are not definite. The article contributes to the question about the adequacy of simulation and presents an attempt to simulate echoscopy and quantitatively estimate simulation uncertainties related to spatial resolution and image contrast of phased transducer scans. Three types of physical phantoms were simulated by appropriate digital phantoms and estimates of three spatial resolution components—axial, lateral, and elevation—are compared with those experimentally measured. It was shown that Field II software modeling of phased array transducer gives from 4.6 to 8.8% average discrepancy of axial and lateral resolution. Beamwidth FWHM in elevation plane is found in around 22% discrepancy from experimental estimate at a focal distance. Modeling tissue by a distribution of scatterers with density 10–15 per resolution volume the simulated image contrast of +12 dB reference inclusion in TMM phantom showed some overestimation of contrast regarding background. This shows the future need for proper fitting of scatterer distribution parameters of a digital phantom to the scattering characteristics of the physical phantom, which are still unspecified. Further research is needed in the modeling and simulation of speckle patterns adequate to the particular type of non-homogeneous tissues what is important in elastography, and quantitative imaging-based on RF signals.

## Figures and Tables

**Figure 1 sensors-21-04420-f001:**
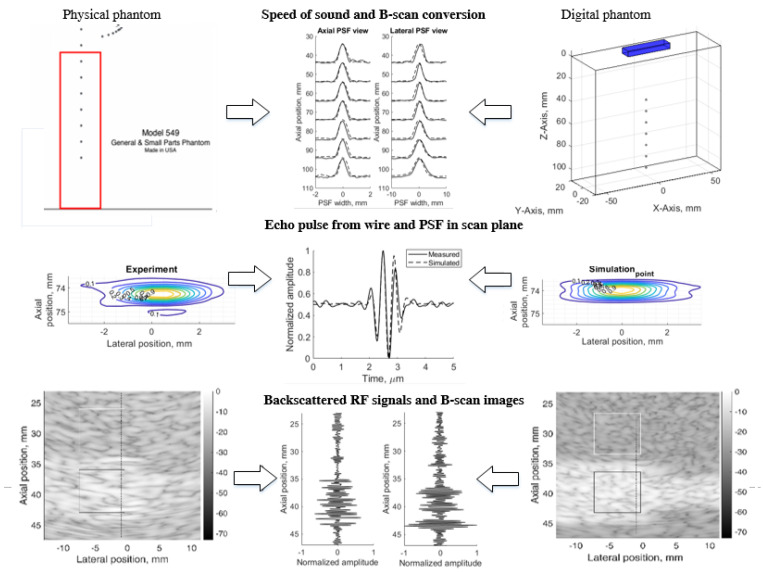
Method of simulation uncertainties evaluation.

**Figure 2 sensors-21-04420-f002:**
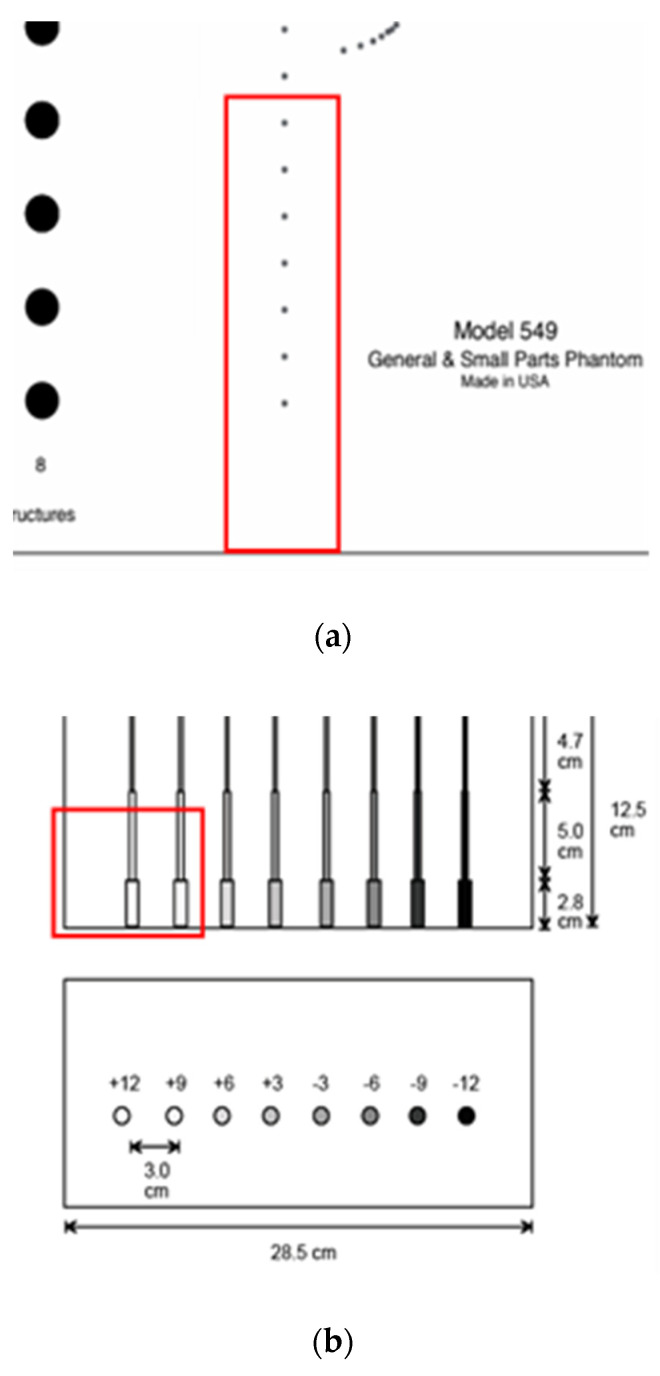
Structure of ATS Laboratories Inc. phantoms (**a**) Model 549 (physical wire phantom), (**b**) Model 532A (physical contrast phantom), and (**c**) Model 538NH (slice thickness phantom) with preselected scanning areas shown. Adopted from [38,39,40].

**Figure 3 sensors-21-04420-f003:**
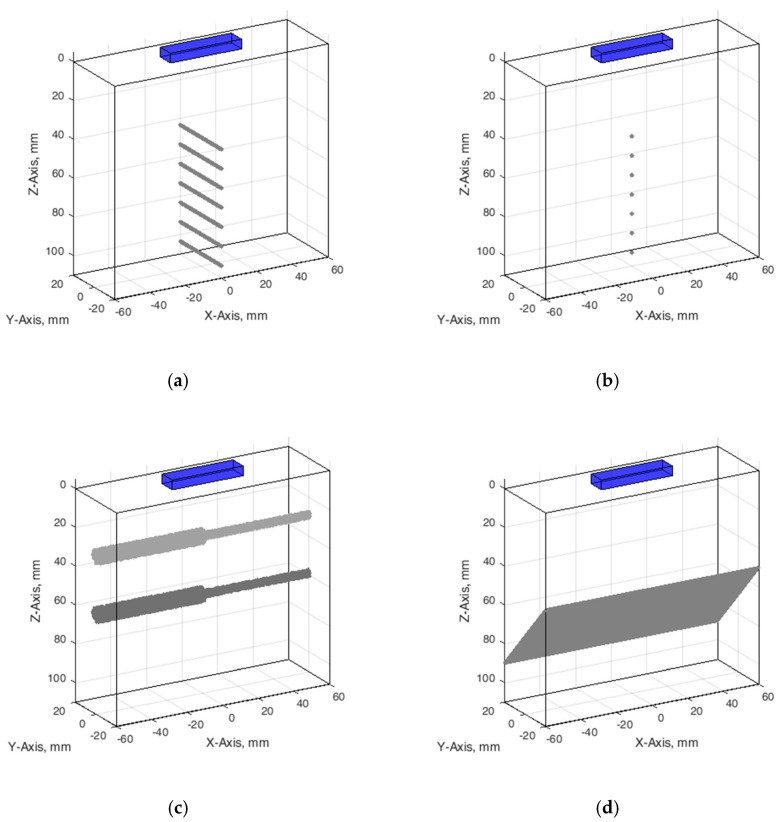
Configuration of digital phantoms: (**a**) wire phantom, (**b**) point phantom, (**c**) contrast phantom, (**d**) slice thickness phantom. The cuboids on the top of digital phantoms have marked a position of the digital ultrasound transducer.

**Figure 4 sensors-21-04420-f004:**
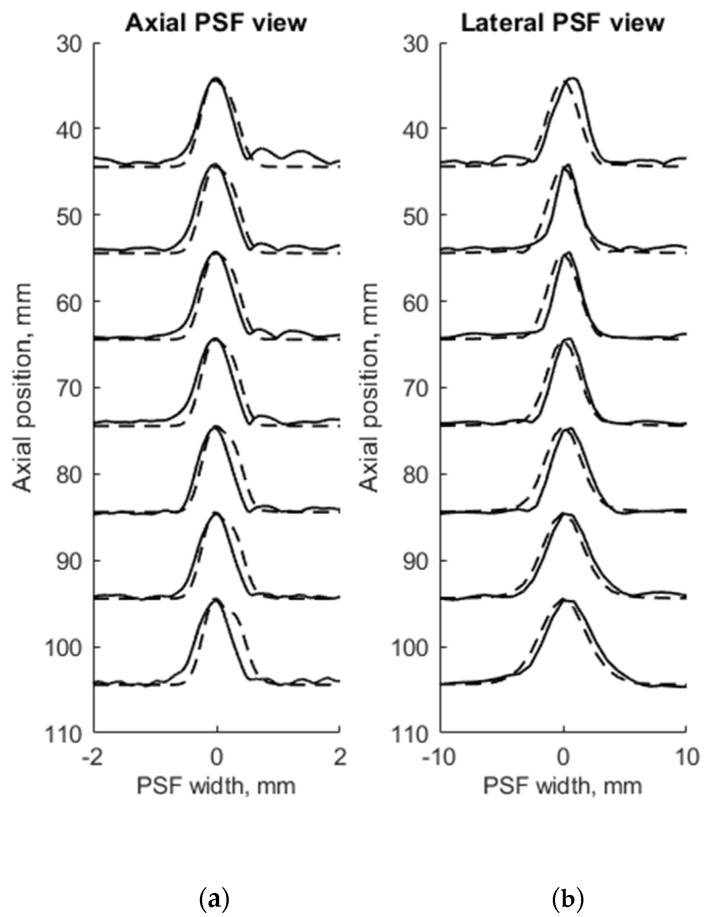
Comparison of the axial (**a**) and lateral (**b**) width of PSF: solid line—experimental echoscopy of physical wire phantom, dashed line—simulation of wire phantom by wires. PSF amplitudes were normalized.

**Figure 5 sensors-21-04420-f005:**
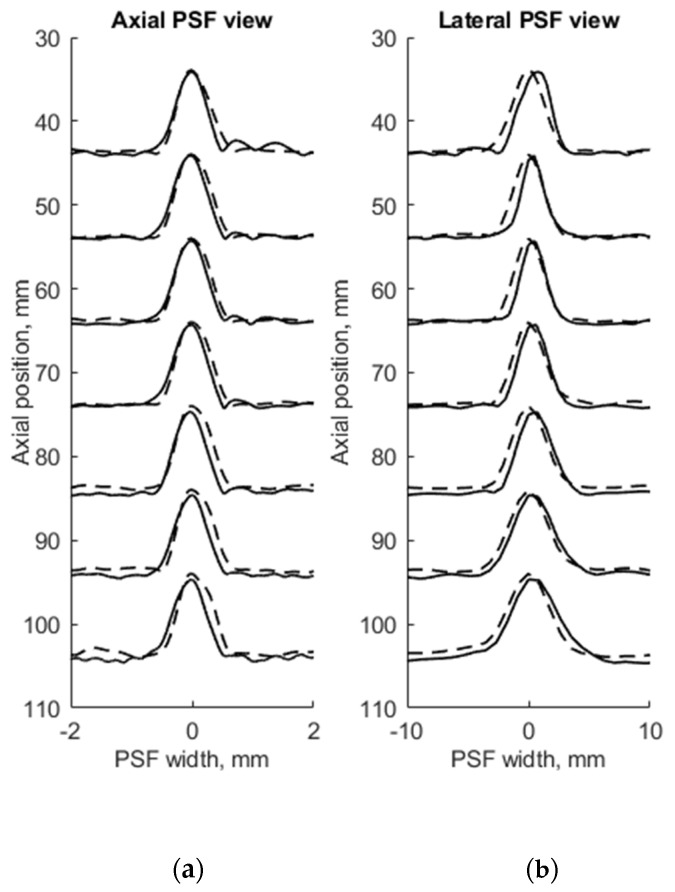
Comparison of axial (**a**) and lateral (**b**) width of PSF: solid line—experimental echoscopy of physical wire phantom, dashed line—simulation of wire phantom by point scatterers. PSF amplitudes were normalized from 0 to 10.

**Figure 6 sensors-21-04420-f006:**
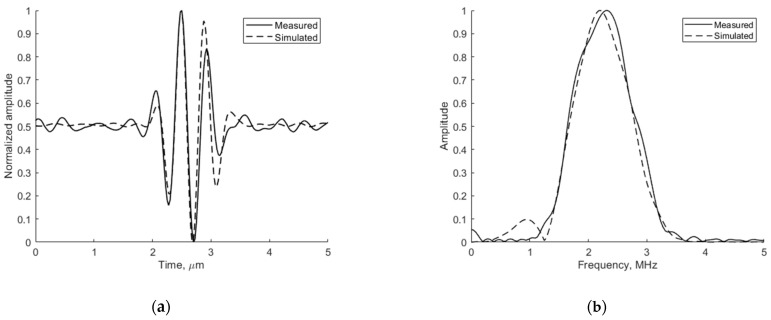
Comparison of measured physical and simulated digital (**a**) pulse-echo waves reflected from wire target (in Model 549) at 74 mm depth (axial view) and (**b**) their spectrum.

**Figure 7 sensors-21-04420-f007:**
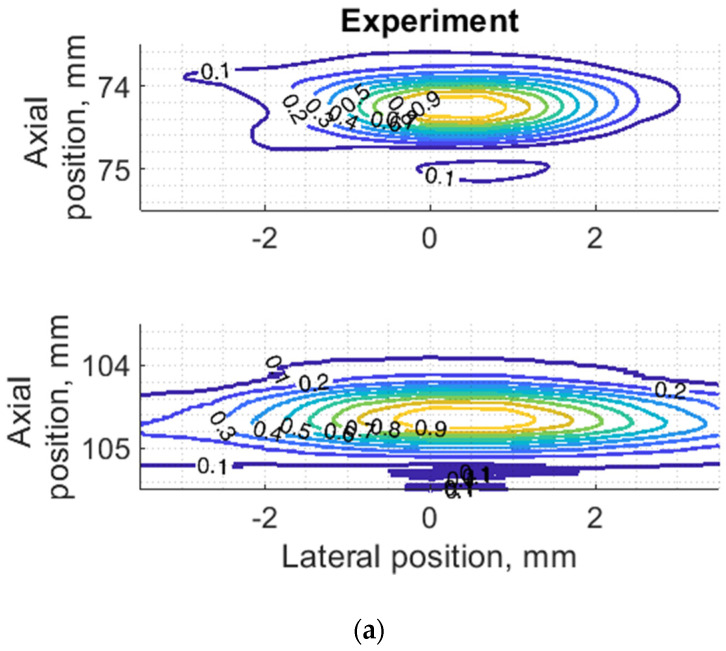
Comparison of PSF using isoline diagrams of RF signals reflected from targets at 74 and 100 mm: (**a**) experimental echoscopy of physical wire phantom, (**b**) simulation of wire phantom as wire scatterers, (**c**) simulation of wire phantom as point scatterers. Isolines presented at 0.1 intervals (from 0.1 to 0.9) of normalized envelopes of RF signals.

**Figure 8 sensors-21-04420-f008:**
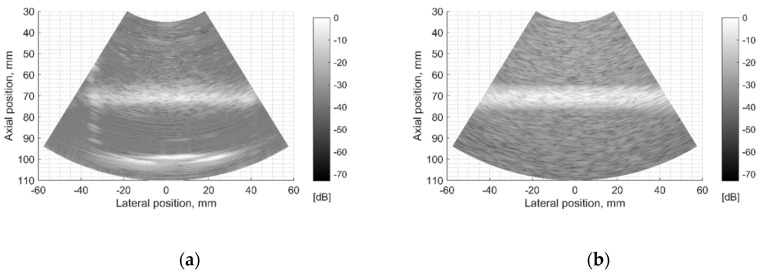
Comparison of slice thickness (elevation resolution) in B-scan images of (**a**) physical Model 538NH phantom and (**b**) digital inclined plane phantom.

**Figure 9 sensors-21-04420-f009:**
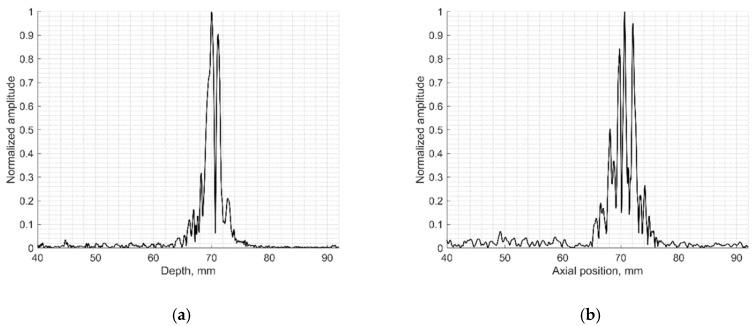
Comparison of signal envelopes backscattered from the inclined plane of slice thickness phantoms: (**a**) physical Model 538NH echoscopy and (**b**) digital phantom simulation.

**Figure 10 sensors-21-04420-f010:**
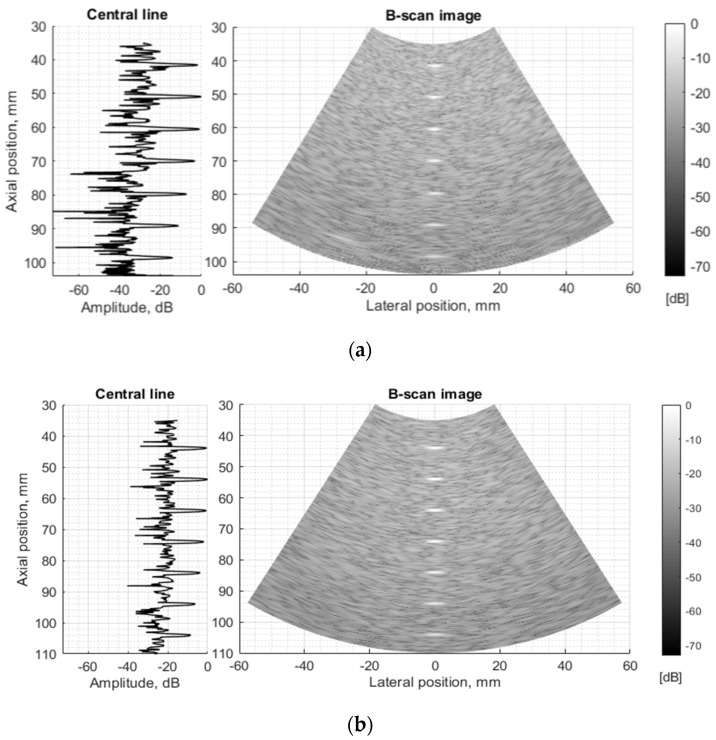
Comparison of envelopes of backscattered signals (**left**) and whole B-scan images (**right**) of (**a**) RF data measured in ATS Model 549 (physical wire phantom) (**b**) simulated RF data of digital wire target phantom.

**Figure 11 sensors-21-04420-f011:**
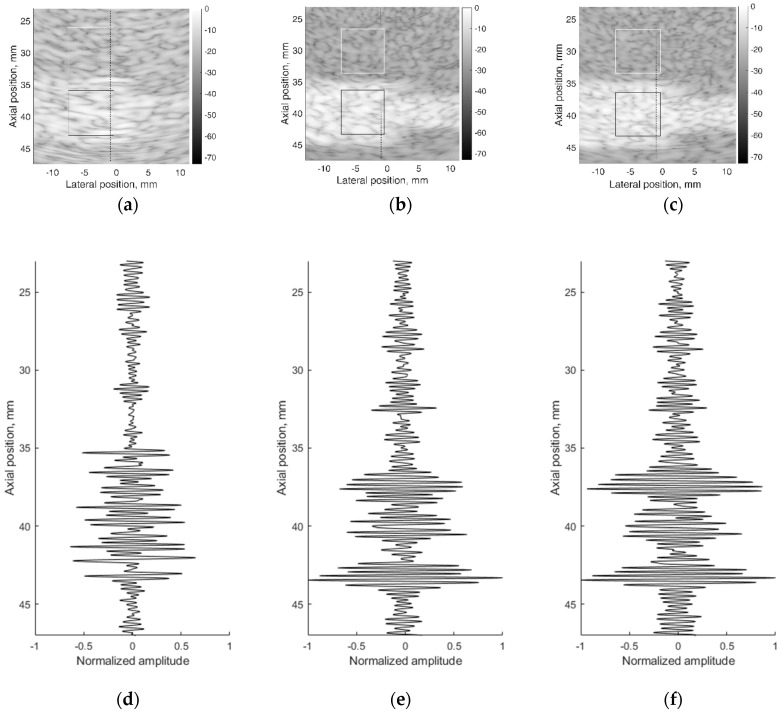
B-scan sub-images of background TMM with contrast inclusion having +12dB stronger scattering: (**a**) physical contrast phantom; (**b**) digital contrast phantom with 10 scatterers per resolution cell (SPRC), (**c**) digital contrast phantom with 15 SPRC and RF signals of corresponding depth ranges along scanning line at lateral (–1) mm position: (**d**) physical contrast phantom, (**e**) digital contrast phantom with 10 SPRC, (**f**) digital contrast phantom with 15 SPRC. The regions (7 × 7 mm) outlined with the white and black lines were taken for B-scan intensity evaluation.

**Table 1 sensors-21-04420-t001:** Parameters of virtual scanning with Field II and settings of transducer and scanner for wire, contrast, and slice thickness phantoms.

Parameters	Field II Values	Transducer and Scanner Values
Array type	Phased array	Phased array
Transducer center frequency	2.52 MHz	2.52 MHz
Number of physical elements	64	64
Apodization (y-direction)	Hanning function	n.a.
Element excitation	Hanning-modulated sinusoid of two cycles	Hanning-modulated sinusoid of two cycles
Height (y-direction) of element	12 mm	12 mm
Width (x-direction) of element	0.24 mm	0.24 mm
Kerf	0.014 mm	0.014 mm
Pitch	0.254 mm	0.254 mm
Elevation lens focus	64 mm	64 mm
Focus	70 mm	70 mm
Element sub-division in x-direction	1	1
Element sub-division in y-direction	13	n.a.
F-number transmit	3	3
F-number receive	3	3
Sampling frequency	40 MHz	40 MHz
Speed of sound	1540 m/s	1540 m/s
Assumed frequency-dependent attenuation	0.5 dB/cm/MHz@ 3.5 MHz	n.a.

n.a.—not available.

**Table 2 sensors-21-04420-t002:** FWHM parameter comparison of the experimental echoscopy of the physical wire phantom and simulation of wire phantom by wires.

Experimental Echoscopy				Simulation				Axial Resolution Relative Error, %	Lateral Resolution Relative Error, %
Axial Position of PSF Peak, mm	Amplitude of PSF Peak, dB	Axial Resolution, mm	Lateral Resolution, mm	Axial Position of PSF Peak, mm	Amplitude of PSF Peak, dB	Axial Resolution, mm	Lateral Resolution, mm		
44.1	−1.81	0.59	3.13	44.4	−0.17	0.65	2.96	11.5	−5.5
54.1	0.00	0.63	2.16	54.4	0.00	0.66	2.89	6.3	33.8
64.3	−0.79	0.62	2.41	64.4	−0.58	0.66	2.89	6.1	20.6
74.3	−3.13	0.63	2.82	74.4	−1.65	0.67	3.09	5.9	9.4
84.7	7.25	0.61	3.36	84.4	−3.26	0.68	3.45	9.4	3.3
94.6	−11.17	0.63	3.94	94.4	−5.04	0.70	3.89	11.1	−1.2
104.7	−14.07	0.65	4.50	104.4	−7.00	0.72	4.40	10.9	−2.2
							**Average, %**	**8.8**	**8.3**

**Table 3 sensors-21-04420-t003:** FWHM parameter comparison of experimental echoscopy of physical wire phantom and simulation of wire phantom by point scatterers.

Experimental Echoscopy				Simulation				Axial Resolution Relative Error, %	Lateral Resolution Relative Error, %
Axial Position of PSF Peak, mm	Amplitude of PSF Peak, dB	Axial Resolution, mm	Lateral Resolution, mm	Axial Position of PSF Peak, mm	Amplitude of PSF Peak, dB	Axial Resolution, mm	Lateral Resolution, mm		
44.1	−1.81	0.59	3.13	43.9	−0.45	0.62	3.06	5.1	−2.2
54.1	0.00	0.63	2.16	54.0	0.00	0.65	2.86	3.2	32.4
64.3	−0.79	0.62	2.41	64.0	−0.69	0.65	2.84	4.8	17.8
74.3	−3.13	0.63	2.82	74.0	−1.86	0.65	3.00	3.2	6.4
84.7	7.25	0.61	3.36	84.0	−3.76	0.66	3.30	8.2	−1.8
94.6	−11.17	0.63	3.94	94.0	−6.31	0.66	3.70	4.8	−6.1
104.7	−14.07	0.65	4.50	104.0	−8.50	0.67	4.04	3.1	−10.2
							**Average, %**	**4.6**	**5.2**

## Data Availability

Not applicable.

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
