# Peer review of "Main Uncertainties in the RF Ultrasound Scanning Simulation of the Standard Ultrasound Phantoms"

_sensors, 2021, doi:10.3390/s21134420_

Round 1

Reviewer 1 Report

Authors have presented quantitative evaluation of major uncertainties in simulation ultrasound imaging specifically for resolution and contrast study. Following are the comments which should be addressed for further improvement of the overall quality of the manuscript.

(1) Please elaborate what are the fitting experiment parameters with relevant references as mentioned on Page 4, line 162.

(2) Which reconstruction algorithm was utilized and why?

(3) Please clarify, when using a wire as target, is it imaged cross-sectional or along the lateral plane of scanning.

(4) What are the properties of digital contrast phantom?

(5) Please explain, what are the uncertainties in digital simulation when the phantom documentation is not available. For example, acoustic properties of the biological tissue is available, however, different literature provide slightly different values. How does this impact the digital simulation? What are the potential ways to resolve this issue?

(6) Which internal scanner parameters were unknown as mentioned on Page 9, line 268, please specify?

(7) Please clarify what is element sub-division. Is it only applicable to phased array? How this parameter would change and affect the simulation for different transducer configuration (i.e. ring array, 2D arrays)

(8) In table 2, the lateral resolution is improved upto 74.4mm, does the transducer has any geometric focal point, please clarify.

(9) In figure 10a, the central line has a dc offset as compared to figure 10b, what is the source of it. Is it due to non-uniformity among the actual elements of the transducer used in physical experiment?

(10) What are the typical materials used as contrast inclusion?

Author Response

Dear reviewer,

Thank you for your work in revising your paper and for giving insightful comments and suggestions for improving our manuscript.

Sincerely,

Authors of the manuscript

Reviewer 2 Report

Apart from the relatively minor issues identified below this paper makes a useful contribution to comparing simulated and measured time-domain ultrasound.  Agreement between simulated and measured results is good.  Proper acknowledgement of the limits of the simulation are required, in particular the use of ideal sources without any matching layer in the simulation.

Major Comments

Lines 289-290     The origin and importance of the XML file needs to be made clear – is this something that is obtained from the scanner?

Lines 293-301     The axis system used needs to be explicitly stated at the start of this paragraph.  In addition, some of the description is specific to the Field II system (e.g. subx and suby) and the meaning needs to be clear without knowledge of the software.

Table 1                How is the elevation lens focus defined and created?

Figures 4 & 5      More details are required in the methods section on how the phantom was positioned and moved to obtain these data 

Line 622              The model used does not include the elements of the transducer or the matching layer.  Increasingly transducers are etched from a single block of material and impedance matching is achieved using multiple layers with different material properties.  Whilst it is outside the scope of the current work, this limitation should be acknowledged and ways in which the modelling could be extended to cover such transducers discussed.

Minor Comments

Line 64                It is not clear on what basis the authors state ‘The most popular simulation packages are …’ and it might be better to say ‘Popular simulation packages include…’

Figure 1               The text on the time series graphs is too small to read.

Line 256              It needs to be made clear whether this is a planar linear array or a curved array

Line 368              ‘Near field’ and ‘far field’ have specific meanings in single element transducers. Either their use here for a focussed ultrasound beam needs to be explained or the wording changed.

Line 429              A reference to the estimation methods relating the widths of the PSF in the lateral and elevation directions is required.

Author Response

(The authors gave the same response as above.)
